# Intraoperative cell salvage: The impact on immune cell numbers

Michelle Roets[1,2]*, David Sturgess[2,3], Thu Tran[4], Maheshi Obeysekera[4], Alexis Perros[4], John-Paul Tung[2,4,5], Robert Flower[4], Andre van Zundert[1,2,5], Melinda Dean[2,4,6]

1 Department of Anaesthesia, Royal Brisbane and Women's Hospital, Brisbane, Queensland, Australia,
2 Faculty of Medicine, University of Queensland, Brisbane, Queensland, Australia, 3 Department of Anaesthesia, Surgical Treatment and Rehabilitation Service, Brisbane, Queensland, Australia, 4 Australian Red Cross Lifeblood, Kelvin Grove, Queensland, Australia, 5 Faculty of Health, Queensland University of Technology, Brisbane, Queensland, Australia, 6 School of Health, University of the Sunshine Coast, Sunshine Coast, Queensland, Australia

* m.roets@uq.edu.au

**Data Availability Statement:** All relevant data are within the paper and its Supporting Information files.

**Funding:** This study was conducted within the Royal Brisbane and Women's hospital (RBWH,

## Abstract

### Background

Patient outcomes are influenced by many confounding factors peri-operatively, including the type of surgery, anaesthesia, transfusion, and immune competence. We have previously demonstrated (*in-vitro*) that compared to allogeneic blood transfusion (ABT), intra-operative cell salvage (ICS) improves immune competence. The peri-operative immune response is complex. Altered or impaired immune responses may predispose patients to develop adverse outcomes (i.e., post-operative wound infection, pneumonia, urinary tract infection etc.) Surgical patients may develop infection, even without the confirmed presence of a definite microbiological pathogen. With all these factors in mind it is important to consider changes in immune cell numbers (and sub-populations) and functional capacity during peri-operative transfusion.

### Methods

In this TRIMICS-Cell (Transfusion Related Immune Modulation and Intraoperative Cell Salvage-Cell numbers) study (n = 17, October 2018-November 2019) we prioritized and analysed peri-operative changes in the number and proportions of immune cell populations and sub-populations (B cells (CD20$^+$), NK (natural killer) cells (CD56$^+$), monocytes (CD14$^+$), T cells (total CD3$^+$ and sub-populations: T helper cells (CD4$^+$), cytotoxic T cells (CD8$^+$), effector T cells (CD4$^+$ CD127$^+$), activated effector T cells (CD4$^+$ CD25$^+$ CD127$^+$) and regulatory T cells (CD4$^+$ CD25$^+$ CD127$^-$)), plasmacytoid dendritic cells (pDC; Lineage$^-$, HLA-DR$^+$, CD11c$^-$, CD123$^+$), classical dendritic cell (cDC) (Lineage$^-$, HLA-DR$^+$, CD11c$^+$), and cDC activation (Lineage$^-$, HLA-DR$^+$, CD11c$^+$), co-stimulatory/adhesion molecules and pDC (CD9$^+$, CD38$^+$, CD80$^+$, CD83$^+$, CD86$^+$, CD123$^+$). Firstly we analysed the whole cohort of study patients and secondly according to the relevant transfusion modality (i.e., three study

Herston, Brisbane, Queensland, Australia). Patient recruitment and sample collection was supported by the intraoperative cell salvage group, the research nursing staff and staff specialist anaesthetists within the anaesthetic department at the RBWH through funding received from the grants mentioned below. Sample analysis occurred at the Australian Red Cross Lifeblood (Herston, Brisbane, Queensland, Australia), who supported the equipment, facilities, and staff, funded in part by the grant below and in part in kind. MR discloses receipt of the following financial support for the research: PhD scholarship grant support [grant number PSc01, $30,000] from the Australian National Blood Authority (NBA, Lyneham, Australian Capital Territory, Australia, online at https://www.blood.gov.au/), administered through the University of Queensland (St Lucia, Brisbane, Queensland, Australia); and from the Australian and New Zealand College of Anaesthetists (ANZCA, Melbourne, Victoria, Australia, online at https://www.anzca.edu.au/); project and scholarship grants ([grant number 18/ 023], $70,000 (2018), $20,000 (2019)), administered through the RBWH and RBWH foundation (Herston, Brisbane, Queensland, Australia). The other authors received no grant funding and the work on the study and manuscript was supported in kind. The funders had no role in study design, data collection and analysis, decision to publish, or preparation of the manuscript.

**Competing interests:** The authors have declared that no competing interests exist.

groups: those who received no transfusion, received ICS only (ICS), or both ICS and allogeneic packed red blood cells (pRBC) (ICS&RBC)), during major orthopaedic surgery.

## Results

For the whole study cohort (all patients), changes in immune cell populations were significant: leucocytes and specifically neutrophils increased post-operatively, returning towards pre-operative numbers by 48h post-operatively (48h), and lymphocytes reduced post-operatively returning to pre-operative numbers by 48h. When considering transfusion modalities, there were no significant peri-operative changes in the no transfusion group for all immune cell populations studied (cell numbers and proportions (%)). Significant changes in cell population numbers (i.e., leucocytes, neutrophils and lymphocytes) were identified in both transfused groups (ICS and ICS&RBC). Considering all patients, changes in immune cell subpopulations (NK cells, monocytes, B cells, T cells and DCs) and functional characteristics (e.g., co-stimulation markers, adhesion, activation, and regulation) were significant perioperatively and when considering transfusion modalities. Interestingly DC numbers and functional capacity were specifically altered following ICS compared to ICS&RBC and pDCs were relatively preserved post-operatively following ICS.

## Conclusion

A transient peri-operative alteration with recovery towards pre-operative numbers by 48h post-surgery was demonstrated for many immune cell populations and sub-populations throughout. Immune cell sub-populations and functional characteristics were similar perioperatively in those who received no transfusion but changed significantly following ICS and ICS&RBC. Interesting changes that require future study are a post-operative monocyte increase in the ICS&RBC group, changes in cDC considering transfusion modalities, and possibly preserved pDC numbers post-operatively following ICS. Future studies to assess changes in immune cell sub-populations, especially during peri-operative transfusion, while considering post-operative adverse outcomes, is recommended.

## Introduction

Allogeneic blood transfusion (ABT) is an essential part of peri-operative (pre-operative, intra-operative and post-operative) patient care, with >118.5 million blood donations collected globally [1], and fresh blood expenditure exceeding $620.7 million per year in Australia alone [2]. Despite improvements in blood safety, there are significant risks associated with ABT (for example incorrect blood component transfused, pulmonary complications, febrile reactions, and infection related adverse outcomes) [3,4]. Alternatively, during intraoperative cell salvage (ICS) transfusion a patient's own blood can be collected, processed and reinfused during surgery [5]. ICS is recommended for specific surgical procedures [6], reduces ABT requirements [7], may provide immunological benefits [8], and improves outcomes (*in vitro* and clinical) [7,9]. This manuscript does not intend to confirm the clinical advantages of ICS over ABT. These advantages have previously been confirmed, by Carless et al. 2010, Meybohm et al. 2016, Roets et al. 2020, and many others [7–10]. However, the detailed mechanism (i.e., specific immunological changes) potentially associated with transfusion-related immune modulation

(TRIM) and improved outcomes following ICS, is unclear. This study aims to further the understanding of relevant changes in immune cell numbers and function during transfusion (and ICS). Information gathered from this study include alternative methods (never used before in TRIM research) and changes that occur peri-operatively. In future study, while working towards the clarification of TRIM, some of the specific changes differentially altered (when considering transfusion groups) in our study should be considered.

Many transfusion related adverse outcomes have immune mediated processes as a key element of their pathogenesis [10,11] and often occur later in the peri-operative journey (i.e., beyond 48 hrs after surgery). One example of an adverse outcome, potentially a consequence of TRIM, is post-operative infection. An estimated 10% of patients develop infection peri-operatively; ten times more frequently than those who experience myocardial ischemia or infarction [12]. Sepsis develops in up to 2% of patients following trauma surgery, with associated increased length of stay (LOS) in hospital (34.1 vs. 7.0 days), in the intensive care unit (ICU) (21.8 vs. 4.7 days) and mortality (23.1% vs. 7.6%) [13,14]. Severe sepsis is the most common cause of mortality in European intensive care patients [15]. The study of peri-operative immune competence and associated exacerbating factors is therefore increasingly important [14].

There is a limited understanding of the immunological processes and mechanisms associated with peri-operative transfusion and infection. The innate immune system is the first line of defence against a foreign pathogen [16]. The adaptive immune system is essential when the innate immune system is ineffective in eliminating infection, to enable the activation of pathogen-specific effector pathways and a longer term immune "memory". T cell responses lead to cellular immunity and B cells mediate antibody production (humoral immunity) [16]. Following binding of major histocompatibility complex (MHC) and co-stimulatory cell-surface molecules I T cells proliferate and differentiate into effector T cells, to act against specific target cells. Effector T cells function through the production of cytokines and the expression of membrane associated effector molecules. Maturation of dendritic cells (DCs) can result in the activation of CD4$^+$ T cells, CD8$^+$ T cells, and natural killer (NK) cells [17]. Different T cell subpopulations have specific functions, for example CD8$^+$ cytotoxic T cells recognise and destroy virus-infected cells, T helper 1 cells promote activation of macrophages and T regulatory cells produce inhibitory cytokines to limit effector responses [17]. Neutrophils play a crucial role in the first defence line against bacterial and fungal pathogens, are the most numerous leucocyte population and the largest subgroup (>95%) of granulocytes [18]. Neutrophils are redistributed (proportional to the extent of the surgery) from the spleen and bone marrow to the circulation and surgical field [19]. The systemic inflammatory response syndrome (SIRS) relates to the human pro-and anti- inflammatory response, following clinical (surgical, infection, traumatic etc.) insult. Typically hyperthermia or hypothermia, tachycardia, tachypnoea and leucocytosis or leucopoenia are considered when clinicians calculate a SIRS score [20]. This score reflects the extend of the relevant inflammatory "insult" and may predict worse outcomes. By studying changes in specific immune cell sub-populations, valuable information could be gained to clarify the pathogenesis associated with TRIM.

Measures to reduce TRIM, and associated adverse outcomes have been implemented including leucocyte-depletion and autotransfusion [4]. However, despite universal leucocyte-depletion, TRIM is still present [21]. Using an *in vitro* transfusion model, we recently demonstrated that exposure to ABT or ICS suppressed monocyte and DC immune responses, with evidence of improved immune competence following ICS [8]. Mossanen *et al.* demonstrated that a) numbers of monocytes reduced immediately post-operatively and increased at day 1 and day 4 post-operatively; and b) that raised leukocytes, neutrophils and intermediate monocytes predicted patients who developed infectious complications following cardiac surgery [22].

Many changes in immune cell numbers and the capacity to respond to infectious insult occur during a traumatic event (i.e., orthopaedic surgery in our study). It is often difficult to separate these processes when studying changes seen during laboratory studies. Surgical patients may have a predisposition to develop infection, even without the presence of a definite pathogen. These changes represent a multifactorial and complex scenario, demonstrating physiological changes related to surgical stress, anaesthesia and the immune consequences of transfusion (including TRIM). To continue our assessment of immune competence following ICS, we focussed our study on peri-operative changes across a panel of important immune cell sub-populations and specifically chose cell sub-populations considering their functional capacity relevant to peri-operative infection risk. We first considered the routinely assessed populations identified in a basic differential full blood count and further studied an extended panel of specialised immune sub-populations, using quantitative flow cytometry. We used this approach to identify immune sub-populations modified by transfusion that could be used as cell targets in larger clinical outcome studies. For the purposes of this manuscript and to simplify readability we will use the word "biomarkers" to describe the immune cell populations and sub-populations (mentioned earlier in this introduction) from here onwards. Transfusion studies powered to consider clinical outcomes such as infection require large samples sizes. A previous meta-analysis by Kim et al. of 21,770 patients, considering allogeneic transfusion, demonstrated an infection rate of 2.88% for transfused patients [23]. The use of a large panel of biomarkers would not be feasible within this large sample size. The aim of our analysis was to evaluate the significance of a focussed panel of biomarkers rather than clinical outcomes. With this smaller sample (17 patients) we were able to evaluate a larger more complex panel of novel biomarkers. To ultimately find robust evidence that can confirm the pathophysiology of TRIM, we first need to find the most valuable biomarkers. These biomarkers can then be used to study clinical outcomes, such as infection in patients, in a large well powered clinical outcome study.

## Materials and methods

### Patient recruitment

Ethics approval was obtained from the Royal Brisbane and Women's Hospital (RBWH) Human Research Ethics Committee (RBWH HREC/17/QRBW/685) and the University of Queensland (UQ) Ethics Committee (2018000297). The trial was registered with the Australian and New Zealand Clinical Trials Registry (ACTRN12618001459213, registered 30/8/2018) [24]. All data collected was de-identified, authors could not identify individual participants after data collection (S1 Table). Written informed consent was obtained from all participants for inclusion in the study.

We recruited and consented elective orthopaedic cases, booked to receive ICS (n = 19) at the RBWH (Brisbane, Queensland, Australia), between October 2018 and November 2019. Surgeons at the RBWH, expecting potential major blood loss during specific procedures, request the availability of ICS (booking). The RBWH cares only for patients older than 12 years of age. Within the observational nature of this study, ongoing recruitment of major elective orthopaedic surgical procedures to the study occurred. There were no exclusion criteria and no patients declined consent. At the RBWH ICS is now part of the standard practise for all elective orthopaedic procedures with major blood loss risk. Blood samples were collected from the arterial line (already in place for the surgery) before and after surgery and by using a standard phlebotomy technique at 48h. This study was observational, no changes to the clinical care of patients were made for the purposes of this study. None of the patients in this study had contraindications to use ICS. Reinfusion of ICS blood was only dictated (as clinically

relevant) by the anaesthetist caring for the patient at the time. The sample sizes were based on previous experience of the authors in this manuscript, of similar preliminary studies. Standard anaesthesia included the use of propofol, fentanyl, midazolam, a muscle relaxant (rocuronium, suxamethonium, cisatracurium, vecuronium) and analgesia (ketamine, lignocaine, oxycodone, local anaesthetic field block, regional or epidural block). To enable ICS the XTRA autotransfusion system (ATS; LivaNova), was used according to a standard operating procedure for orthopaedic cases [8].

### Blood sample collection and analysis

Peripheral blood samples (10 mL each into ethylenediaminetetraacetic acid (EDTA) phlebotomy tubes (Becton Dickinson (BD), Oxford, England)) were collected from the patient preoperatively, immediately post-operatively and at 48h post-operatively (48h) (Fig 1). A total of 19 patients had blood samples available for assessment. One patient only received packed red blood cells (pRBC) and was therefore excluded from subsequent analyses. One patient was excluded for whom we did not have a 48h sample (the patient was discharged from hospital before the sample was due). Responses were compared across the three study groups: those who had 1) no transfusion (n = 4), 2) ICS only (ICS, n = 9) and 3) both ICS and RBC (ICS&RBC, n = 4) (total number of patients analysed were n = 17).

**Measurements used in this study.** During clinical patient care and in preparation for anaesthesia during major surgery, a 5-part differential full blood count (Neutrophils (40–60%), lymphocytes (B cells and T cells, 20–40%), monocytes (2–8%), eosinophils (1–4%) and basophils (0.5–1%)) is routinely performed in hospitals [25]. During this study, an automated 3-way differential cell count (which includes the total leucocyte count, neutrophils (y %), mid-sized cells (MID; monocytes, eosinophils, basophils; x %) and lymphocytes (z %)) (Cell-Dyn Emerald Haematology Analyser, Abbott, Chicago, USA), detailed immune cell sub-populations (Trucount analysis) and cell ratio (e.g., lymphocyte-monocyte ratio (LMR)) were assessed at three time points (previously mentioned). Haematocrit and platelet counts were also assessed. The Lymphocyte-Monocyte ratio (LMR) was calculated using Emerald data (i.e., lymphocyte number / MID number).

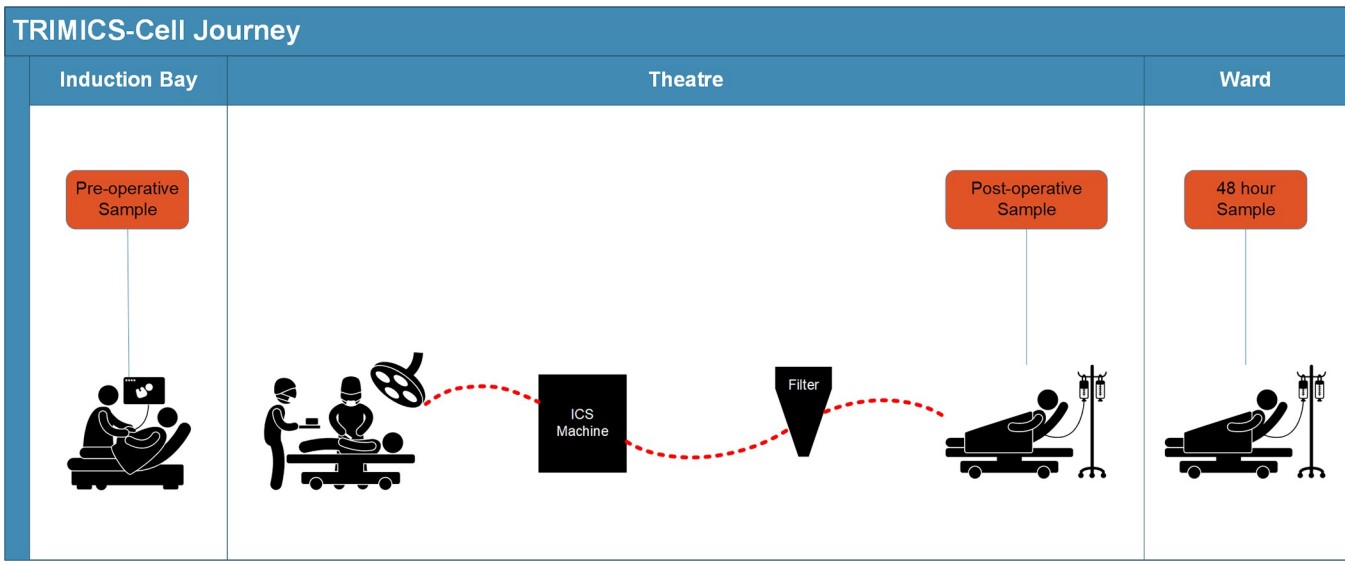

**Fig 1. TRIMICS-cell patient journey (pre-operative-, post-operative-, 48h post-operative samples).**

**Quantification of leucocyte sub-populations and DC specific co-stimulatory and adhesion molecules.** To enable a more detailed comparison of the immune cell profile, quantitative flow cytometry using Trucount tubes (BD Biosciences) was used to determine the absolute counts of the following leucocyte populations in whole blood: B cells ($CD20^+$), NK cells ($CD56^+$), monocytes ($CD14^+$), T cells (total $CD3^+$ and sub-populations: T helper cells ($CD4^+$), Cytotoxic T cells ($CD8^+$), Effector T cells ($CD4^+$ $CD127^+$), activated effector T cells ($CD4^+$ $CD25^+$ $CD127^+$) and Regulatory T cells ($CD4^+$ $CD25^+$ $CD127^-$)), Plasmacytoid DCs (pDC; Lineage$^-$, HLA-DR$^+$, $CD11c^-$, $CD123^+$), Classical DC (Lineage$^-$, HLA-DR$^+$, $CD11c^+$), and Classical DC activation (Lineage$^-$, HLA-DR$^+$, $CD11c^+$), Co-stimulatory/Adhesion Molecules and pDC ($CD9^+$, $CD38^+$, $CD80^+$, $CD83^+$, $CD86^+$, $CD123^+$) at each different time point in the cell salvage procedure. Trucount tubes and all fluorescent labelled monoclonal antibodies (Table 1) were from BD Biosciences (San Jose, USA). Gating strategy was based on a previous study of cardiac surgery patients [26]. The Trucount procedure was performed according to the manufacturer's instructions and absolute cell count was calculated by multiplying the number of cellular events divided by the number of bead events, the lot specific bead number divided by the test volume [26].

**Flow cytometry.** A 3-laser FACSCanto™ II flow cytometer with FACS Diva (both BD Biosciences, San Jose, USA) was used. The absolute count of leucocyte populations, using Trucount tubes, was analysed using FCS Express V6 (De Novo Software, Pasadena, USA).

## Statistical analysis

Firstly, we considered the full patient cohort and assessed changes in leucocyte populations (from basic full blood count) and specific immune cell sub-populations. Secondly, we compared transfusion groups and assessed leucocyte populations (from basic full blood count) and specific immune cell sub-populations. We compared each sample with every other sample, at various time points. A repeated measures ANOVA with Friedmans test and Dunn's multiple comparison test was used to count and analyse multiple immune cell populations; firstly, considering all patients and then for each study group at each time point, using GraphPad Prism

**Table 1. Panel of monoclonal antibodies for quantification of immune cell sub-populations, adapted with permission [26].**

| B Cell, NK Cell, Monocyte Panel | T Cell Sub-population Panel | DC Panel |
|---|---|---|
| CD56 FITC (clone NCAM 16.2) | CD4 FITC (clone RPA-T4) | Lineage Negative Cocktail 2 FITC (CD3 (clone SK7), CD14 (clone MΦP9), CD20 (clone L27), CD19 (clone SJ25C1), CD56 (clone NCAM16)) |
| CD14 V500 (clone M5E2) | CD8 V500 (clone RPA-T8) | CD34 FITC (clone 8G12) |
| CD45 PerCP (clone 2D1) | CD45 PerCP (clone 2D1) | CD45 PerCP (clone 2D1) |
| CD20 APC (clone L27) | CD3 APC-H7 (clone SK7) | HLA-DR V450 (clone L243) |
| | CD25 V450 (clone M-A251) | CD11c APC (clone S-HCL-3) |
| | CD127 PE (clone HIL-7R-M21) | CDx* PE |

***Co-stimulatory/Adhesion Molecules and pDC**: CD9 (clone M-L13), CD38 (clone HIT2), CD40 (clone 5C3), CD80 (clone L307.4), CD83 (clone HB15e), CD86 (clone FUN-1), CD123 (clone 9F5). All antibodies were conjugated with PE.

7, GraphPad software. ANOVA *P*<0.05 was considered statistically significant and Dunn's multiple comparison test indicated in figures as *P<0.05, **P<0.01, ***P<0.001.

## Results

### General demographics

The study group consisted of 56% male (n = 10) and 44% female (n = 8) patients. The mean age was 45 years ± 22 (SD) (range 18–88) on the first day of surgery.

### Changes in leucocyte populations during orthopaedic surgery

Firstly, we assessed the changes in immune cell populations (determined by automated basic full blood count) for all study patients, throughout the surgical journey. Changes in haematocrit (*P* = 0.001, Fig 2A) and platelet counts (*P*<0.001, Fig 2B) were also recorded. Total leucocyte numbers increased post-operatively and remained higher than pre-operative levels at 48h

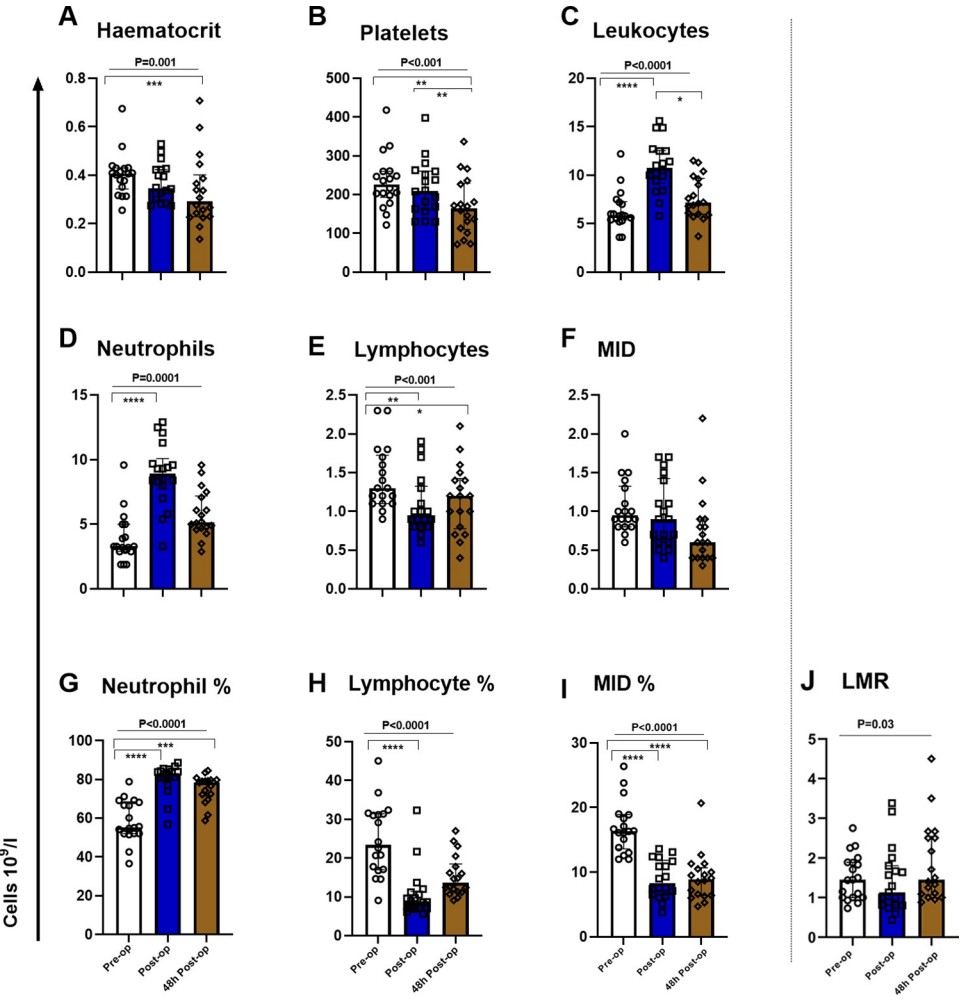

**Fig 2. Three-part differential cell count of peri-operative changes in the main leucocyte populations for all patients.** Automated haematology measurement. Y axis unit 109/l. X axis, three time points; pre-op (pre-operatively), post-op (post-operatively) and at 48h (48h post-operatively). ANOVA (Friedman test), indicated by horizontal bar with *P* value above and Dunn's multiple comparison test: *P<0.05, **P<0.01, ***P<0.001, ****P<0.0001. MID, Mid-range cells (monocytes, basophils, eosinophils); LMR, Lymphocyte-Monocyte-Ratio.

($P<0.0001$, Fig 2C). Neutrophil numbers increased post-operatively ($P = 0.0001$) and remained higher than pre-operatively (a trend) (Fig 2D). Lymphocyte numbers reduced post-operatively with recovery evident, however remained significantly lower than pre-operatively at 48h ($P<0.001$, Fig 2E). Changes in the number of MID cells were not statistically significant (Fig 2F). Proportional (%) changes in leucocyte populations were assessed. Neutrophil (%) increased significantly post-operatively and reduced at 48h, remaining much higher than pre-operatively ($P<0.0001$, Fig 2G). Lymphocyte (%) was significantly reduced post-operatively ($P<0.0001$, Fig 2H). MID % was significantly reduced post-operatively and remained less than half of pre-operative numbers by 48h ($P<0.0001$, Fig 2I). Considering all patients, most immune cell populations (except neutrophils) had reduced numbers at the post-operative and 48h time points.

The value of LMR to predict cancer recurrence and survival after cancer surgery has been studied and an association found between a lower LMR ($<2.85$) and a higher tumor grade, higher age and worse overall survival [27]. To consider whether LMR would be of value when studying TRIM peri-operatively we compared LMR at relevant time points in our study. LMR significantly changed peri-operatively, decreasing post-operatively before returning to pre-operative numbers by 48h ($P = 0.03$, Fig 2J).

## Changes in leucocyte sub-populations during orthopaedic surgery, comparing transfusion groups

We further assessed whether immune cell populations (numbers and proportions) were impacted differently when different transfusion modalities were used (i.e., no transfusion, ICS or ICS&RBC; Fig 3). There were no significant peri-operative changes in the no transfusion group for all cell populations (numbers and proportions (%)). Leucocyte numbers increased post-operatively following ICS ($P<0.0001$) and ICS&RBC ($P = 0.042$, Fig 3A). These changes seem more pronounced following ICS. Neutrophils increased post-operatively ($P<0.0001$) and remained above pre-operative levels at 48h (trend), following ICS ($P<0.0001$) and ICS&RBC ($P = 0.042$, Fig 3B). Changes were more pronounced in patients who received ICS only. Lymphocytes were reduced post-operatively following ICS ($P = 0.012$) and ICS&RBC ($P = 0.009$, Fig 3C). MID numbers were significantly lower in the ICS&RBC group peri-operatively ($P = 0.037$, Fig 3D). The proportion of neutrophils (%) increased post-operatively following ICS ($P<0.0001$) and ICS&RBC ($P = 0.005$, Fig 3E). Lymphocytes (%) reduced post-operatively following ICS ($P<0.0001$) and ICS&RBC ($P = 0.005$), trending towards pre-operative numbers by 48h (Fig 3F). The proportion of MID cells were significantly reduced post-operatively and at 48h, following ICS ($P<0.001$), Fig 3G.

## Changes in leucocyte sub-populations during orthopaedic surgery

In a detailed analysis, that included all study patients, we assessed the peri-operative impact to immune cell sub-population numbers and considered the impact this would have to the capacity to respond to infectious insult. Sub-populations included in our study were NK cells, monocytes, B cells, CD3+-, CD8+-, CD4+-T cell sub-populations including T regulatory cells, and DCs (Fig 4L–4R). Specific DC sub-populations were included to enable evaluation of various immune responses (e.g., expression of activation, co-stimulation, proliferation etc.) and to assess the capacity to respond to potential infectious insult during surgery.

Peri-operative changes in NK cell numbers were significant ($P = 0.034$) but not pronounced (Fig 4A). Interestingly monocytes ($P = 0.003$, Fig 4B) and B cells ($P = 0.009$, Fig 4C) increased between the post-operative and 48h time points. A reduction in CD3+ T cells was seen post-operatively ($P<0.001$, Fig 4D). Cytotoxic T cells (CD8+) were significantly reduced post-

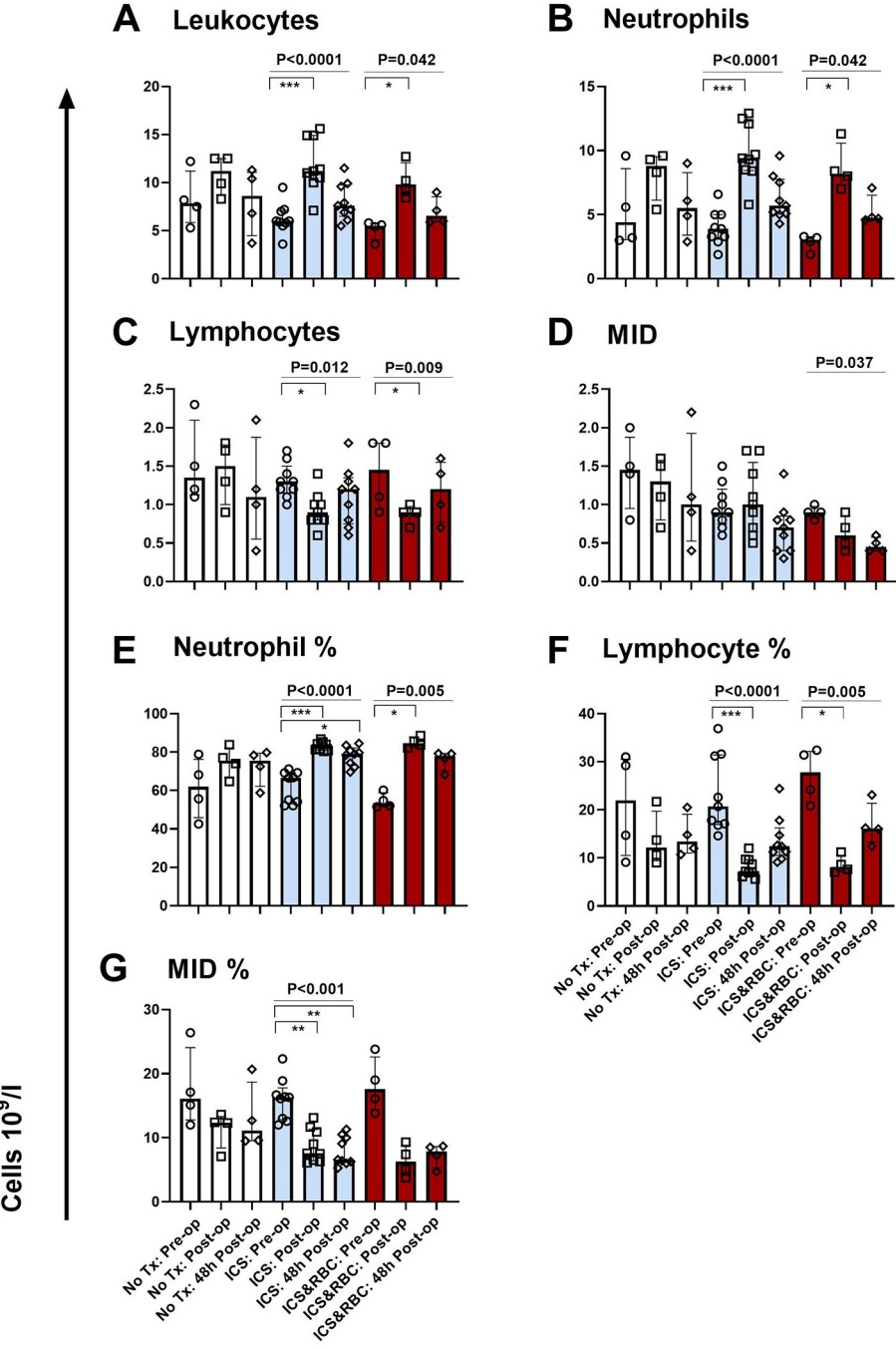

**Fig 3. Three-part differential cell count peri-operative changes in individual immune cell populations, comparing transfusion groups.** Y axis, cell numbers and proportions, unit $10^9$/l. X axis, three time points (pre-operatively (pre-op), post-operatively (post-op) and at 48h post-operatively (48h)) and study groups (no transfusion, intraoperative cell salvage (ICS), intraoperative cell salvage and red blood cells (ICS&RBC)). ANOVA (Friedman test), indicated by horizontal bar with P value above and Dunn's multiple comparison test: *P<0.05, **P<0.01, ***P<0.001, ****P<0.0001. MID, Mid-range cells (monocytes, basophils, eosinophils); LMR, Lymphocyte-Monocyte-Ratio; %, proportion.

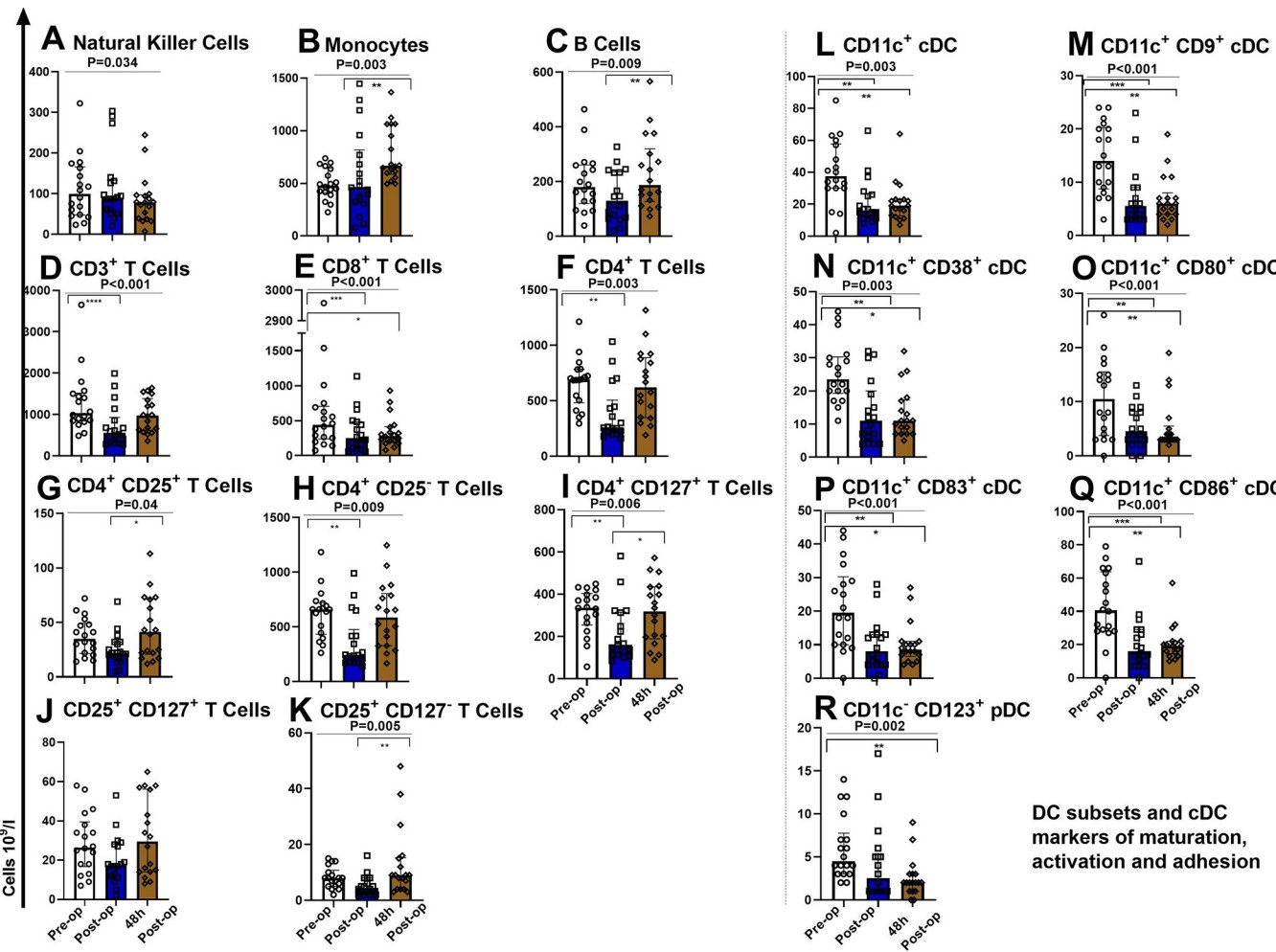

**Fig 4. Analysis of peri-operative changes in leucocyte sub-populations, including all study patients.** Y axis unit $10^9$/L. X axis, three time points; pre-op (pre-operatively), post-op (post-operatively) and at 48h (48h post-operatively). ANOVA (Friedman test), indicated by horizontal bar with *P* value above and Dunn's multiple comparison test: *P<0.05, **P<0.01, ***P<0.001, ****P<0.0001.

operatively and at 48h (*P*<0.001, Fig 4E). In our study CD4+ CD25+ T cells changed, significantly increasing between the post-operative and 48h time points (*P* = 0.04, Fig 4G). CD4+ CD25- T Cells were reduced post-operatively (*P* = 0.009, Fig 4H). CD4+ CD127+ effector T cells were reduced post-operatively and increased at 48h (*P* = 0.006) (Fig 4I). CD25+ CD127+ T cells did not change significantly (Fig 4J). Regulatory T cells have immune suppressive properties [28]. CD25+ CD127- T cells (T regulatory cells) reduced between the post-operative and 48h time points (*P* = 0.005, Fig 4K).

In addition we studied the number of cells and expression of co-stimulatory and adhesion molecules considering classical DCs (cDCs) and plasmacytoid DCs (pDCs) [29] (Fig 4L–4R). cDC numbers were suppressed post-operatively (more than 50%) and remained suppressed at 48h, not returning to pre-operative numbers (*P* = 0.003, Fig 4L). The number of cDCs expressing co-stimulatory molecules CD9+ (P<0.001), CD38+ (*P* = 0.003), CD80+ (P<0.001), CD83+ (P<0.001) and CD86+ (P<0.001), decreased post-operatively, remaining low at 48h (Fig 4M–4Q). pDC were supressed post-operatively and remained low at 48h (*P* = 0.002) (Fig 4R).

## Changes in leucocyte sub-populations during orthopaedic surgery, according to transfusion group

To evaluate the impact of different transfusion modalities, we assessed the changes in immune cell sub-population numbers according to transfusion groups (Fig 5). No statistically significant changes occurred peri-operatively for those who had no transfusion (Fig 5A–5R). NK cell numbers changed peri-operatively following ICS ($P = 0.048$, Fig 5A). Significant peri-operative changes occurred in monocyte numbers following ICS&RBC, resulting in increased numbers at 48h ($P = 0.042$, Fig 5B). B cells were suppressed post-operatively and increased above pre-operative numbers at 48h, following ICS ($P < 0.001$, Fig 5C).

When considering T cells, CD3+ T cells ($P < 0.001$, Fig 5D) and CD8+ T cells ($P = 0.001$, Fig 5E) were significantly reduced post-operatively following ICS. Significant changes occurred in CD4+ T cell sub-population numbers post-operatively, reduced following ICS ($P = 0.01$, Fig 5F) and following ICS&RBC ($P = 0.042$, Fig 5F), not returning to pre-operative levels following ICS (trend). Considering CD4+ cell numbers, CD4+ CD25+ were increased at

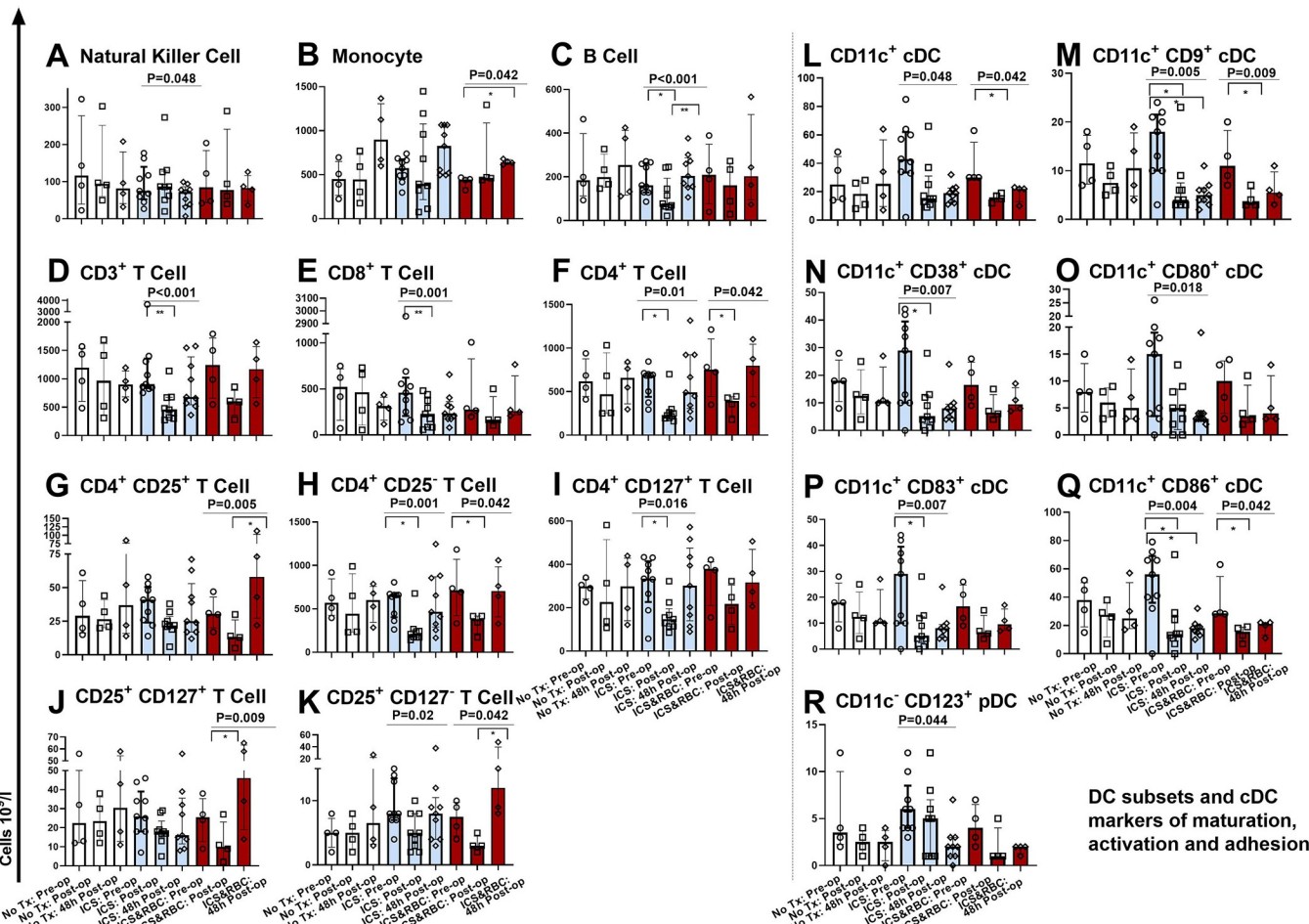

**Fig 5. Analysis of peri-operative changes in leucocyte sub-populations, comparing transfusion groups.** Y axis, cell numbers and proportions, unit $10^9$/l. X axis, three time points (pre-operatively (pre-op), post-operatively (post-op) and at 48h post-operatively (48h)) and study groups (no transfusion, intraoperative cell salvage (ICS), intraoperative cell salvage and red blood cells (ICS&RBC)). ANOVA (Friedman test), indicated by horizontal bar with P value above and *-**** Dunn's multiple comparison test: *P<0.05, **P<0.01, ***P<0.001, ****P<0.0001. No Tx, no transfusion; ICS, intraoperative cell savage; ICS&RBC, intraoperative cell savage and allogeneic packed red blood cells.

48h following ICS&RBC ($P$ = 0.005, Fig 4G). Changes in, CD4$^+$ CD25$^-$ T cells (as counterpart of CD4$^+$ CD25$^+$) were significant and pronounced; suppressed post-operatively for both ICS ($P$ = 0.001) and ICS&RBC ($P$ = 0.0042, Fig 5H). CD4$^+$ CD127$^+$ T cells, markers for the cell's activation state (i.e., effector cell activation), significantly reduced post-operatively following ICS ($P$ = 0.016, Fig 5I) [30]. Significant reduction in CD25$^+$ CD127$^+$ T cells occurred following ICS&RBC post-operatively ($P$ = 0.009), with a pronounced increase (trend) at 48h (Fig 5J). Changes in CD25$^+$ CD127$^-$ T cells (regulatory T cells) were significant following ICS peri-operatively (P = 0.02) and at 48h following ICS&RBC ($P$ = 0.042), resulting in a pronounced increase at 48h following ICS&RBC (Fig 5K).

DCs were assessed within our panel, because they act as an important bridge between the innate and adaptive immune systems. The peri-operative reduction in cDC was significant following ICS ($P$ = 0.048) and mostly at 48h following ICS&RBC ($P$ = 0.042, Fig 5L). Significant peri-operative suppression occurred in CD9$^+$ cDC following ICS post-operatively and at 48h ($P$ = 0.005) and following ICS&RBC post-operatively ($P$ = 0.009, Fig 5M). Following ICS, CD38$^+$ cDC was supressed post-operatively and at 48h ($P$ = 0.012, Fig 5N), CD80$^+$ peri-operatively ($P$ = 0.018, Fig 5O) and CD83$^+$ post-operatively ($P$ = 0.007; Fig 5P). cDCs expressing CD86$^+$ were significantly suppressed post-operatively and at 48h in the ICS group ($P$ = 0.004, Fig 5Q), and post-operatively in the ICS&ABT group ($P$ = 0.042, Fig 5P). Significant peri-operative changes were seen following ICS in pDC ($P$ = 0.044) (Fig 5R).

## Discussion

Earlier, it has been demonstrated that many clinical adverse outcomes, believed to be associated with TRIM (for example post-operative infection), are reduced by ICS [7]. The patho-physiology of TRIM is however unclear. Finding biomarkers differentially altered by different autologous (i.e., ICS) or allogeneic transfusion modalities is therefore essential. Clinicians use "biomarkers" (for example leucocytosis >12*10$^9$/l or leucopoenia <4*10$^9$/l) to define SIRS and predict adverse outcomes in critically ill patients (for example in multi-organ dysfunction syndrome). The large panel of biomarkers included in this study were considered to evaluate changes in immune biology, that occur during major surgery where transfusion could be expected. Specific biomarkers were chosen to reflect various roles of immune cells (and the clinical relevance), for example CD4$^+$ sub-populations (effector/capacity to drive the immune response), B cells (ensure a sustained immune response), and other immune cells that are involved in activation, adhesion, regulation etc. [31,32]. Figures noting peri-operative changes in haematocrit and platelet numbers were included to provide some clinical context (in the background) of this study. However, as our study aimed to assess changes in immune cells, detailed analysis of platelet numbers and haematocrit were outside the objectives of this study.

We assessed changes in the numbers of leukocyte populations. These changes were firstly assessed while considering all the patients in the study and thereafter while considering the transfusion groups. The significant differences in "all the patients" graphs were likely due to changes in the "transfused groups" (i.e., ICS and ICS&RBC). For all the patients in the study, leucocytes increased post-operatively and returned towards pre-operative numbers by 48h. Leukocytes increased post-operatively following ICS and ICS&RBC. Neutrophils provide crucial innate immune functions during severe infections; they are recruited from the bone marrow to the circulation [19]. For all the patients in our study, neutrophil numbers increased post-operatively. When considering study groups, neutrophil numbers increased post-operatively regardless of transfusion modality (i.e., ICS or ICS&RBC). A pronounced and significant increase in post-operative neutrophil numbers may have clinical implications. Immunological homeostasis (i.e., a balance between pro- and anti-inflammatory mediators and a balanced

cell-mediated response) is important to resist infectious insults and promote recovery from surgical trauma. Both an excessive or persistent inflammatory response and/or impaired cell-mediated immunity after surgery (i.e., "immune paralysis"), increase a patient's risk to develop infection [14]. Neutrophil function is however complex. Therefore, we are cautious to make conclusions regarding potential associated clinical outcomes from these results. For all the patients in our study, lymphocytes decreased post-operatively and returned towards pre-operative numbers by 48h. Lymphocytes decreased post-operatively following ICS and ICS&RBC. Lymphopenia resulting from "stress-induced apoptosis", cellular exhaustion, desensitization, and down-regulation, was previously associated with severe infection, and increased mortality [33–35]. In our study (when considering transfusion groups), the numbers of lymphocytes were not significantly reduced at 48h compared to pre-operatively. Failure to normalize lymphopenia following trauma may be associated with increased mortality [33]. The MID population (numbers and %) in our study included mostly monocytes, with a smaller proportion of eosinophils and basophils. Within this category monocytes represent an important immune cell population, relevant when studying peri-operative adverse outcomes. For the study population as a whole, changes in MID (%) were significant, reduced post-operatively, increasing at 48h (still lower than pre-operatively). When considering transfusion groups, changes in MID % were significant post-operatively and at 48h, reduced for ICS. Changes in monocytes, specifically related to time points within the surgical journey are important. Both, prolonged monocyte deactivation (functional impairment immediately after surgery) as well as increased monocyte numbers post-operatively (beyond day 1) were previously associated with adverse outcomes [22,36]. MID % (rather than MID cell numbers) may be a valuable measurement in future clinical outcome studies. LMR was not different following ICS compared to RBC&ICS. Importantly, in patients who did not receive a transfusion, no statistically significant peri-operative changes were demonstrated for all cell populations studied (cell numbers and proportions (%)).

There is a paucity of evidence specific to changes in immune cell sub-populations during peri-operative transfusion [33]. To find specific biomarkers of value for future study, we did a more detailed analysis of the immune profile. Sub-populations of immune cells, relevant to cell function were assessed. Again, we first considered changes in immune profile for all the patients in our study, and then according to transfusion group. Our further analysis based on transfusion modality demonstrated that there were no statistically significant peri-operative changes in any of the cell sub-populations studied (cell numbers and proportions (%)), for patients who did not receive a transfusion. However, we did find changes in immune cell sub-populations in patients who received ICS or ICS&RBC.

For all the patients in our study NK cells reduced post-operatively and at 48h. When considering transfusion groups, NK cells increased slightly peri-operatively following ICS. NK cells (CD56+) function through the release of cytotoxic proteins to destroy cancer cells and virus infected cells, secrete IFN-γ and TNF-α that act on macrophages and dendritic cells and regulate (activate or inhibit) the response against specific antigens [37,38]. A reduction in NK cell numbers, as a marker of cell-mediated immunity, may be associated with infection related adverse outcomes [39]. For all the study patients, there was a significant increase in monocyte numbers at 48h. During the transfusion group assessment, when considering ICS&RBC, monocyte numbers were significantly increased at 48h (compared to pre-operative numbers). Changes in monocyte numbers were not significant following ICS. Monocytes are phagocytic, among the first responders during infectious insult, and enable recruitment and activation of other immune cells [40]. Post-operative impaired monocyte function was previously associated with a susceptibility to develop infection related complications [19]. A combination of increased monocyte numbers and impaired CD11c+ cDC expression was previously

associated with immune suppression and increased risk to develop opportunistic infections [41]. The significant increase of monocyte numbers at 48h following ICS&RBC is an important finding to consider in future study; to assess the association between monocyte populations, peri-operative transfusion and post-operative adverse outcomes.

B cells recognise antigens directly and differentiate into antibody-secreting plasma cells or memory B cells. B cell numbers were reduced between the post-operative and 48h time points when considering all patients. When considering transfused groups, B cells were reduced post-operatively but recovered to above pre-operative numbers at 48h following ICS. A reduction immediately post-operatively may represent a transient suppression or suggest that an inadequate B cell response may lead to an increased risk to develop subsequent infection. A better understanding of the B cell activation or phenotype may improve our understanding of the consequences (clinical relevance) of this outcome.

As both T cell and B cells were reduced following surgery for total knee replacement and CD3$^+$, CD4$^+$- and CD8$^+$ T cells transiently reduced during trauma, orthopaedic surgery, and ABT we also investigated changes in T cells in our study [39,42]. In our study CD4$^+$ T cell sub-population numbers were reduced immediately post-operatively considering all patients and similarly following ICS and ICS&RBC. Post-operative reduction in CD3$^+$ T cells, may suggest impaired defences against pathogens and a susceptibility to develop infection [43]. The impaired cytotoxic ability (CD8$^+$ T cells) considering all patients and following ICS, may suggest impaired protective immunity against malignant cells and intracellular pathogens (viral infection) [44]. Furthermore, Long et al. demonstrated through an *in vitro* model that exposure to stored pRBC reduced T cell and B cell proliferation, but that this was eliminated when fresh blood was used [45]. We did not demonstrate a similar protection during clinical patient care. CD4$^+$ T cell and C8$^+$ T cell numbers were significantly depressed post-operatively following ICS, despite these patients receiving fresh blood.

Regulatory T cells, important in the immune response during injury, regulate immune responses and ensure homeostasis [30,46]. Their role in protecting against infectious diseases may be beneficial or detrimental depending on the type of infection or timing of activation. The impact of these cells on the immune response is dependent on the balance between effector and regulatory cells [30]. By studying the expression of CD25$^+$ on CD4$^+$ T cells regulatory and effector type cells can be distinguished [30]. When considering all the study patients and when considering ICS&RBC, CD4+ CD25+ numbers were increased at 48h. Considering all the study patients, CD4$^+$ CD127$^+$ numbers, marking effector cell activation in gated data where there are no transcription factors, were reduced post-operatively and increased at 48h. CD4$^+$ CD127$^+$ numbers were reduced post-operatively following ICS. Further detailed analysis demonstrated no clear alteration of the CD4$^+$ phenotype towards a change in the distribution of expression for CD127 (CD4$^+$ CD127$^-$ and $^+$) relatively to CD4$^+$ T cells.

DCs (lineage$^-$ HLA-DR$^+$ cDC) present antigens to T cells and B cells and are specialized and uniquely placed to initiate and regulate the adaptive immune response [47]. We considered the expression of co-stimulatory and adhesion molecules on DCs. Most sub-populations were significantly suppressed post-operatively and remained low at 48h. Considering all the patients, the DC changes in our study reflect both a reduction in actual cell numbers (CD11$^+$ cDC) and the capacity to respond and to stimulate T cells (CD9$^+$ cDC). Potential implications include antigen presentation (cDC), protection against bacterial infection and possibly tumor immunity [29]. When considering transfusion groups, cell numbers (CD11$^+$ cDC) were reduced following RBC&ICS post-operatively. The capacity to respond (CD9$^+$ cDC) was reduced following ICS and RBC&ICS post-operatively; but remained significantly reduced following ICS at 48h. DCs are mobilised into the circulation in response to surgical stress [32], but it was unclear in our study whether the reduction in cDCs and pDCs was because they

were lost, consumed or just not recruited into the circulation. Mature DCs play a role in the activation of CD4[+] and CD8[+] cells, the activation and priming of NK cells, T cell survival, development of T helper cells, and B cell activation [48]. The study of DC numbers (differencially altered when considering ICS and ICS&RBC) may therefore provide valuable insights when considering a patient's risk to develop transfusion related infection.

By studying the presence of cell surface molecules such as CD38[+] (adhesion and signaling), CD80[+], CD83[+] (T and B cell activation) and CD86[+] costimulation, we are able to assess the patient's DC ability to respond to infectious insult (LPS), the ability to support (through antigen presentation and co-stimulation) the generation of CD4[+] and CD8[+] cells and to evaluate T and B cell activation and proliferation [49–52]. In our study the sustained suppression of CD38[+], CD80[+], CD83[+] cDC considering all patients, and specifically following ICS, post-operatively and at 48h, suggest impairment in adhesion and signalling of T and B cells. When considering all patients and transfusion groups the numbers of DC and the capacity to respond to infectious insult were significantly reduced. DC function is intrinsically specific to each DC sub-population [29]. When we considered the phenotype of these DCs the impact following transfusion did not relate to a particular sub-population of cDC. However we did find both lower numbers and consistently supressed activation (CD11c[+] cDC) and adhesion (CD9[+] cDC) throught. When differentiating between cDC and pDC, a reduction was clear in both these sub-populations peri-operatively in our study. Supressed capacity to respond to infection could be expected [48]. Interestingly pDC were significantly changed following ICS at 48h; possibly preserved immediately post-operatively. This finding may have important consequences for patients when responding to infection and should be considered in future clinical outcome studies.

This study was successful in quantifying a panel of immune cell sub-populations as potential biomarkers to study when considering clinical outcomes comparing ICS and ICS&ABT (or potentially ABT) in future. While the small sample size limits the clinical conclusions that can be drawn, we have established in principle that numbers of immune cells changed peri-operatively, and these changes were specifically associated with different transfusion modalities. An important finding is the fact that changes to immune cell populations (and sub-populations) were not evident in patients who did not receive transfusion, but significant in those who did receive ICS and ICS&RBC. Relationships between immune cell numbers and clinical outcomes, in other circumstances, were previously demonstrated. Biomarkers identified in this study would be useful when considering peri-operative transfusion related outcomes, and when comparing differences in outcomes following ICS and ABT. It would be feasible to consider a small panel of relevant biomarkers (identified in our study) during adequately powered clinical outcome studies, to compare ICS and ABT. It would be valuable for future studies to further delineate changes in immune cell profile due to transfusion modality, including a RBC only group (if clinical and ethical considerations allow).

## Conclusion

When considering all the study patients, peri-operative changes in immune cell populations and sub-populations were significant. There were no significant peri-operative changes, for all immune cell populations studied (cell numbers and proportions (%)), in surgical patients who did not receive transfusion. However, when considering transfused groups (ICS&RBC and ICS), significant changes were evident in population numbers and functional capacity (e.g., co-stimulation markers, adhesion, activation, and regulation). Areas of interest that would be valuable to study when comparing ICS vs RBC in future include: CD4[+] CD25[+], CD4[+] CD25[-], CD4[+] CD127[+] and CD25[+] CD127[-] T cells, DC cell numbers (CD11[+] cDC), capacity to respond

(CD9⁺ cDC) and pDCs. Further study of changes in immune cell sub-populations, within the peri-operative journey, and in association with different transfusion modalities, may provide valuable insights in the evaluation and prediction of infection related adverse outcomes, peri-operatively in the future [32].

## Supporting information

**S1 Table. Data file.** TRIMICS-Cell study data.
(XLSX)

## Acknowledgments

The ICS group (Anaesthetic department, Royal Brisbane and Women's Hospital, Brisbane, Queensland, Australia): Mr David Cullingham (Dip (AT)), Ms Trisha Bushell (Dip (AT)), Mr Warick Fawkes (Dip (AT)), Ms Russini Stapleton (Dip (AT)), Ms Yves Long (Dip (AT)), Mr Barry Elliott (Dip (AT)), Ms Cassie Hohnke (Dip (AT)), Ms Lee Elliott (Dip (AT)), Ms Kym Webster (Dip (AT)), Ms Vicki Swaine (Dip (AT), Director Anaesthetic Healthcare Practitioners) provided ICS and helped with data collection. A special thank you to Peter Freeman (Dip (AT)), who tirelessly supported our ICS service for decades. Also at the Royal Brisbane and Women's Hospital (Brisbane, Queensland, Australia): Thank you to Ms Sue Williams (B App Sc (Med Tech) Supervising Scientist, Transfusion Haematology, Central Laboratory Pathology Qld.), Janelle T Toombes (BA, registered nurse, Cancer Care Services Nursing administration, Transfusion Clinical Nurse Consultant (CNC)), Natasha Keary (Cancer Care Services Nursing administration, Transfusion CNC) and Dr John Rowell (Director of haematology (at the time of data collection) and Pathology Queensland). For information technology support, to collect and analyse data, we would like to thank Charles Cheung (BbiomedSc (Hons). PG Dip. MSc, Information Technology) Anaesthetic department, RBWH, Brisbane, Queensland, Australia).

## Author Contributions

**Conceptualization:** Michelle Roets, David Sturgess, John-Paul Tung, Robert Flower, Melinda Dean.

**Data curation:** Michelle Roets, Thu Tran, Maheshi Obeysekera, Alexis Perros, Melinda Dean.

**Formal analysis:** Michelle Roets, Melinda Dean.

**Investigation:** Thu Tran, Maheshi Obeysekera, Alexis Perros, Melinda Dean.

**Resources:** John-Paul Tung, Robert Flower, Melinda Dean.

**Supervision:** David Sturgess, Robert Flower, Andre van Zundert, Melinda Dean.

**Writing – original draft:** Michelle Roets, David Sturgess, Thu Tran, Maheshi Obeysekera, Alexis Perros, John-Paul Tung, Robert Flower, Andre van Zundert, Melinda Dean.

**Writing – review & editing:** Michelle Roets, David Sturgess, Thu Tran, Maheshi Obeysekera, Alexis Perros, John-Paul Tung, Robert Flower, Andre van Zundert, Melinda Dean.

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
