## [Decision Letter · Decision Letter 0]

25 May 2023

PONE-D-23-10523Intraoperative cell salvage: The impact on immune cell numbersPLOS ONE

Dear Dr. Roets,

Thank you for submitting your manuscript to PLOS ONE. After careful consideration, we feel that it has merit but does not fully meet PLOS ONE’s publication criteria as it currently stands. Therefore, we invite you to submit a revised version of the manuscript that addresses the points raised during the review process.

Please submit your revised manuscript by Jul 09 2023 11:59PM.  If you will need more time than this to complete your revisions, please reply to this message or contact the journal office at plosone@plos.org. Please include the following items when submitting your revised manuscript:A rebuttal letter that responds to each point raised by the academic editor and reviewer(s). You should upload this letter as a separate file labeled 'Response to Reviewers'.A marked-up copy of your manuscript that highlights changes made to the original version. You should upload this as a separate file labeled 'Revised Manuscript with Track Changes'.An unmarked version of your revised paper without tracked changes. You should upload this as a separate file labeled 'Manuscript'.

We look forward to receiving your revised manuscript.

Kind regards,

Afsheen Raza, PhD

Academic Editor

PLOS ONE

“*This study was supported in part by a grant from the ANZCA Foundation, Australian and New Zealand College of Anaesthetists*.”  

The ICS group (Anaesthetic department, Royal Brisbane and Women’s Hospital, Brisbane, Queensland, Australia): Mr David Cullingham (Dip (AT)), Ms Trisha Bushell (Dip (AT)), Mr Warick Fawkes (Dip (AT)), Ms Russini Stapleton (Dip (AT)), Ms Yves Long (Dip (AT)), Mr Barry Elliott (Dip (AT)), Ms Cassie Hohnke (Dip (AT)), Ms Lee Elliott (Dip (AT)), Ms Kym Webster (Dip (AT)), Ms Vicki Swaine (Dip (AT), Director Anaesthetic Healthcare Practitioners) provided ICS and helped with data collection. A special thank you to Peter Freeman (Dip (AT)), who tirelessly supported our ICS service for decades. Also at the Royal Brisbane and Women’s Hospital (Brisbane, Queensland, Australia): Thank you to Ms Sue Williams (B App Sc (Med Tech) Supervising Scientist, Transfusion Haematology, Central Laboratory Pathology Qld.), Janelle T Toombes (BA, registered nurse, Cancer Care Services Nursing administration, Transfusion Clinical Nurse Consultant (CNC)), Natasha Keary (Cancer Care Services Nursing administration, Transfusion CNC) and Dr John Rowell (Director of haematology (at the time of data collection) and Pathology Queensland). For information technology support, to collect and analyse data, we would like to thank Charles Cheung (BbiomedSc (Hons). PG Dip. MSc, Information Technology) Anaesthetic department, RBWH, Brisbane, Queensland, Australia).  Collaborative study with Australian Red Cross Lifeblood (permission to disclose); Australian governments fund Australian Red Cross Lifeblood to provide blood, blood products and services to the Australian community.”

“This study was conducted within the Royal Brisbane and Women’s hospital (RBWH, Herston, Brisbane, Queensland, Australia). Patient recruitment and sample collection was supported by the intraoperative cell savage group, the research nursing staff and staff specialist anaesthetists within the anaesthetic department at the RBWH through funding received from the grants mentioned below.

Sample analysis occurred at the Australian Red Cross Lifeblood (Herston, Brisbane, Queensland, Australia), who supported the equipment, facilities, and staff, funded in part by the grant below and in part in kind.

MR discloses receipt of the following financial support for the research: PhD scholarship grant support [grant number PSc01, $30,000] from the Australian National Blood Authority (NBA, Lyneham, Australian Capital Territory, Australia), online at https://www.blood.gov.au/, administered through the University of Queensland (St Lucia, Brisbane, Queensland, Australia); and from the Australian and New Zealand College of Anaesthetists (ANZCA, Melbourne, Victoria, Australia), online at https://www.anzca.edu.au/; project and scholarship grants ([grant number 18/023], $70,000 (2018), $20,000 (2019)), administered through the RBWH and RBWH foundation (Herston, Brisbane, Queensland, Australia).

The other authors received no grant funding and the work on the study and manuscript was supported in kind.

Reviewers' comments:

Reviewer's Responses to Questions

**Comments to the Author**

1. Is the manuscript technically sound, and do the data support the conclusions?

Reviewer #1: No

Reviewer #2: Partly

2. Has the statistical analysis been performed appropriately and rigorously? 

Reviewer #1: No

Reviewer #2: I Don't Know

3. Have the authors made all data underlying the findings in their manuscript fully available?

Reviewer #1: Yes

Reviewer #2: Yes

4. Is the manuscript presented in an intelligible fashion and written in standard English?

Reviewer #1: Yes

Reviewer #2: No

5. Review Comments to the Author

Reviewer #1: Comments:

The dynamic change of immune status during the peri-operative period is an interesting topic. Altered or impaired immune responses may predispose patients to develop adverse outcomes. In this research, the authors analysed the peri-operative changes in the number and proportions of immune cell populations and sub-populations in patients with or without transfusion. The difference between the ICS and ICS&RBC groups was also compared. Although the study had some clinical implications, the study design had defects and some questions should be addressed.

Major concerns:

1.Transfusion-related immune modulation (TRIM) was an important adverse outcome of transfusion, which might lead to post-operative infection. In this study, although the dynamic changes of immune cell populations and sub-populations were analyzed, the correlation between the change of immune status and patient prognosis like infection was not studied, which seriously weakened the clinical value of the study.

2.If the authors tried to explore the advantage of ICS over the ABT, the difference between the ICS and RBC groups should be compared, rather than the comparison between the ICS and ICS&RBC groups. The results of immune cell populations and sub-populations for ICS and ICS&RBC groups were quite similar, which could not draw any meaningful conclusion.

3.As the authors claimed, the change of immune status during the peri-operative period could be affected by many factors. In this research, only elective orthopaedic cases were recruited. However, the detailed clinical information was missing. Were there differences, like bleeding volume, operation time and scope, etc.，between these groups? Did the patients who received ICS or ICS&RBC suffered more trauma or bleeding during the operation? In addition, would the patients show similar change of immune status when receiving other surgeries?

4.For the results of all patients and patients divided by groups, there was an obvious inconsistency. Patients without transfusion did not show significant change of immune cell populations and sub-populations, while those with transfusion did. Due to the limited number of patients without transfusion, the result of all patients showed difference. This conclusion was not rigorous. If recruiting more patients without transfusion, the result of all patients might be opposite.

5.The number of patients recruited was 19 showed in the Abstract. However, in the part of Patient recruitment, there was 5 patients of no transfusion, 8 patients of ICS, 4 patients of ICS&RBC and 1 patient excluded (5+8+4+1=18). Moreover, in the figures, there were 4 dots for the no transfusion group and 9 dots for the ICS group.

Minor concerns:

1.No FCM gating figure or gating strategy of the result.

2.In the figures, the arrow on the left indicated that the ordinate was measured in 10^9 cells/L. However, the results of cell populations measured in percentage were mixed. In addition, for cell sub-populations in figure 4 and 5, the the units for the cell account should be mistake.

3.Regulatory T cells were defined as CD4+CD25+CD127- cells in the part of Materials and methods. However, in the part of Results, CD4+CD25+ T cells were regarded as Tregs (Line 260, Page 13).

4.The FCM channel for detection of Co-stimulatory/Adhesion Molecules and pDC was unknown.

Reviewer #2: I am a clinician with experience of ICS and welcome that research is being performed in this area. Concerns relating to TRIM are hugely important though knowledge of mechanism and risk are not well known. I see this very detailed work has been done to shed light on the topic and raise further questions and identify topics for investigation. I found it hugely complicated and confusing to understand what the actual effects were of the intervention and the potential effect upon immune function as a result. I note the n of each group is small and then there is the problem that any issue found from ABT ( which is the "alternative" to ICS ) is confused by the fact that the ABT group contains ICS. The TRIM issue is suspected to be dose dependent ( Horvath et al - donor transfusion and peri-op infection ). I could not identify whether the "dose or volume" of transfusion product was considered.

I would have considered that there needed to be three distinct groups ( control, ABT, ICS) and hence the effects are more clear as to whether it is transfusion alone or donor blood or ICS that affects the white cell populations.

I tried hard with multiple re-reads to really understand what was going on and what the inference or points were that the authors wish to portray to the reader. If the intended reader were to be a laboratory transfusion/immunology expert, this may be more clear. For the clinician, does the work give a clear indication that ICS has an immunological effect, should patients be informed that TRIM occurs with ICS , is this different to the TRIM of donor blood?

Would a benefit in message and ease of interpretation be achieved if the number of cell types were reduced in this paper, even considering a second paper to describe additional cell types.

Specific remarks

42- received- spelt incorrectly

288 - unit 109/L - unsure what this refers to

287-297 I could not understand this section.

307- in the results section, this sentence appeared to be out of context and such a phrase should have been either in the introduction of discussion section.

Clearly a huge amount of very focused and specialised study and extensively researched discussion. But, due to the complexity, I fear the message and inference/influence may be obscured. I would find it difficult to include a "message " from this paper in any presentations or papers that I would write on this subject, and this is probably due to the complexity and comprehensive nature of the text.

I also think that an ICS only and Transfusion Only group would be required, but understand the ethical issues behind this ( if there is a C/I for using ICS, this may already echo a difference in the immune status of the patient ).

6. PLOS authors have the option to publish the peer review history of their article (what does this mean?). If published, this will include your full peer review and any attached files.

Reviewer #1: No

Reviewer #2: **Yes: **Dr Craig Carroll

---

## [Author Response · Author response to Decision Letter 0]

6 Jul 2023

Specific comments in reply to comments from the editor and reviewers are uploaded within "Response to reviewers" document. Different colours are used to ensure ease of reading within the uploaded document. This information is copied here below:

Please find included in this revision: changed style (as required) and newly labelled figures. We did upload our figures in the “Conversion Engine (PACE) digital diagnostic tool”, to address PLOS ONE's style requirements, including those for file naming. Please do let me know if I can improve anything.

Please remove any funding-related text from the manuscript and let us know how you would like to update your Funding Statement. 

Thank you. Funding information was removed from the acknowledgement section and placed within the funding statement instead. We would like to make changes to our financial disclosure. Please find the updated statement here in the cover letter:

“This study was conducted within the Royal Brisbane and Women’s hospital (RBWH, Herston, Brisbane, Queensland, Australia). Patient recruitment and sample collection was supported by the intraoperative cell salvage group, the research nursing staff and staff specialist anaesthetists within the anaesthetic department at the RBWH, through funding received from the grants mentioned below. Sample analysis occurred at the Australian Red Cross Lifeblood (Herston, Brisbane, Queensland, Australia), who supported the equipment, facilities, and staff, funded in part by the grant below and in part in kind. This was a collaborative study with Australian Red Cross Lifeblood. 

MR discloses receipt of the following financial support for the research: PhD scholarship grant support [grant number PSc01, $30,000] from the Australian National Blood Authority (NBA, Lyneham, Australian Capital Territory, Australia), online at https://www.blood.gov.au/, administered through the University of Queensland (St Lucia, Brisbane, Queensland, Australia); and from the Australian and New Zealand College of Anaesthetists (ANZCA, Melbourne, Victoria, Australia), online at https://www.anzca.edu.au/; project and scholarship grants ([grant number 18/023], $70,000 (2018), $20,000 (2019)), administered through the RBWH and RBWH foundation (Herston, Brisbane, Queensland, Australia). 

Statements required by ARCLB and ANZCA: “Australian governments fund Australian Red Cross Lifeblood to provide blood, blood products and services to the Australian community”; and “This study was supported in part by a grant from the ANZCA Foundation, Australian and New Zealand College of Anaesthetists.” 

The other authors received no grant funding and the work on the study and manuscript was supported in kind. The funders had no role in study design, data collection and analysis, decision to publish, or preparation of the manuscript.”

3. “We note that you have stated that you will provide repository information for your data at acceptance. Should your manuscript be accepted for publication, we will hold it until you provide the relevant accession numbers or DOIs necessary to access your data. If you wish to make changes to your Data Availability statement, please describe these changes in your cover letter and we will update your Data Availability statement to reflect the information you provide.”

Thank you. Please find the data file (excel) attached to this revision. 

Caption for supporting information within the manuscript is as follows: 

S1 File. Data file. TRIMICS-Cell study data. (S1_File.pdf)

4. Please include captions for your Supporting Information files at the end of your manuscript, and update any in-text citations to match accordingly.

We added supporting information files (mentioned before) at the end of our manuscript and updated in-text citations to match accordingly.

Reviewers' comments:

1. Is the manuscript technically sound, and do the data support the conclusions?

Reviewer #1: No

Reviewer #2: Partly

Please find below the relevant changes to the manuscript and clarification of the use of data to support our conclusions.

2. Has the statistical analysis been performed appropriately and rigorously? 

Reviewer #1: No

Reviewer #2: I Don't Know

Please find below the relevant changes to the manuscript and clarification of the use of data and rational behind statistical analysis, to support conclusions.

3. Have the authors made all data underlying the findings in their manuscript fully available?

The study data will be made available with the revised manuscript as supporting information, cited within the manuscript, and noted in the supplemental section.

4. Is the manuscript presented in an intelligible fashion and written in standard English?

Reviewer #1: Yes

Reviewer #2: No

Please find below the relevant changes to the manuscript, clarification of the use of data and rational behind statistical analysis, to support conclusions.

Thank you to reviewers 1 and 2 for their comments. We acknowledge it is a complex and comprehensive study. We have taken steps to refine the core information, to improve readability and ensure the key outcomes are clearer. Please find relevant changes within the revised manuscript and a major rewrite of the discussion section, with the aim to refine the purpose of the study and key messages. Additional literature references were removed. 

We considered a large panel of immune cells and changes in immune cell sub-populations (now simply mentioned as “biomarkers” in the revised manuscript). We found many significant but complex changes during surgery and transfusion.

This is a novel study; detailed changes in immune cell sub-population numbers have never been assessed when considering ICS. Some of the specific changes in immune responses (as identified in our study), were previously associated with clinical outcomes (during different clinical procedures and circumstances). To put these changes into perspective and make sense of the potential clinical relevance of each change (previously confirmed), we relate each observed biological change to relevant clinical aspects identified within an extensive clinical literature review. These changes require further study; considering the potential clinical implications associated with each change.

Standard methods previously used to find the cause or describe the mechanism of TRIM (and outcomes following ICS) did so far not provide all the answers. We therefore aimed to investigate alternative assessments of immune responses during transfusion. This is the first study of its kind. We identify important peri-operative changes in immune cell responses, never identified before. Furthermore, we identify a new approach, a new method to potentially use in future when studying the potential causes of TRIM; and when (in future) studying peri-operative adverse outcomes following transfusion (specifically after ‘orthopaedic surgery that require transfusion’). We do not have all the answers yet, this work describes the progress towards finding answers.

Reviewer #1: Comments:

The dynamic change of immune status during the peri-operative period is an interesting topic. Altered or impaired immune responses may predispose patients to develop adverse outcomes. In this research, the authors analysed the peri-operative changes in the number and proportions of immune cell populations and sub-populations in patients with or without transfusion. The difference between the ICS and ICS&RBC groups was also compared. Although the study had some clinical implications, the study design had defects and some questions should be addressed.

Major concerns:

1. Reviewer 1 comment: “Transfusion-related immune modulation (TRIM) was an important adverse outcome of transfusion, which might lead to post-operative infection. In this study, although the dynamic changes of immune cell populations and sub-populations were analyzed, the correlation between the change of immune status and patient prognosis like infection was not studied, which seriously weakened the clinical value of the study.”

Thank you to reviewer 1 for this comment. This is an important aspect that need clarification. Consideration of clinical outcomes (patient prognosis or infection risk) was beyond the scope of this study. Instead, we aimed to evaluate changes in immune biology during peri-operative transfusion. 

Yet there is a gap in the literature to inform us about which biomarkers would be of most value during a clinical outcome study. We specifically chose the panel of immune cells to direct future study towards potentially associated clinical outcomes (identified through extensive literature review). Specific and selective biomarkers (modulated by transfusion) can then be applied to a larger clinical outcome study in future.

Added to manuscript: “For the purposes of this manuscript and to simplify readability we will use the word “biomarkers” to describe the immune cell populations and sub-populations (mentioned earlier in this introduction) from here onwards. Transfusion studies powered to consider clinical outcomes such as infection require large samples sizes. A previous meta-analysis by Kim et al. of 21,770 patients, considering allogeneic transfusion, demonstrated an infection rate of 2.88% for transfused patients [1]. The use of a large panel of biomarkers would not be feasible within this large sample size. The aim of our analysis was to evaluate the significance of a focussed panel of biomarkers rather than clinical outcomes. With this smaller sample (17 patients) we were able to evaluate a larger more complex panel of novel biomarkers. To ultimately find robust evidence that can confirm the pathophysiology of TRIM, we first need to find the most valuable biomarkers. These biomarkers can then be used to study clinical outcomes, such as infection in patients, in a large well powered clinical outcome study.”

2. Reviewer 1 comment: “If the authors tried to explore the advantage of ICS over the ABT, the difference between the ICS and RBC groups should be compared, rather than the comparison between the ICS and ICS&RBC groups. The results of immune cell populations and sub-populations for ICS and ICS&RBC groups were quite similar, which could not draw any meaningful conclusion.”

Thank you for this important comment. We did not aim to demonstrate the advantages of ICS over RBC. It was previously confirmed, through meta-analysis (and extensive further study over decades), that ICS provide benefits to patients when considering infection related outcomes [2, 3]. We hope to provide evidence of differentially modulated biomarkers potentially associated with the outcomes we see in clinical outcome studies (during ICS); and to find biomarkers with potential to use in future TRIM research. We agree that we now know (considering the results of this study) that many biomarkers in our panel were not specifically altered by ICS or RBC&ICS and would likely not be useful biomarkers during future TRIM (and ICS) research. However, we did find significant differences in some sub-populations (Fig 5). To conclude, we did therefore define a smaller panel of immune cells with potential, to use in future during adequately powered clinical outcome studies, when working towards finding the mechanism of TRIM.

Most clinicians would currently use ICS if it was available. We believe it would no longer be ethically acceptable to randomise patients to a “no ICS” group for research purposes, if ICS was available. Clinically relevant groups, as in this observational study, are most commonly ICS and ICS&RBC (reflected in the fact that there was only one case that received ‘only RBC’ in our observational study). There are no longer contraindications to ICS in this group of patients. We aimed to find differences in biology between clinically relevant groups.

Added to the manuscript: “This manuscript does not intend to confirm the clinical advantages of ICS over ABT. These advantages have previously been confirmed (by Carless et al. 2010, Meybohm et al. 2016, Roets et al. 2020, and many others) [2-5]. However, the detailed mechanism (i.e., specific immunological changes) potentially associated with transfusion-related immune modulation (TRIM) and improved outcomes following ICS, is unclear. This study aims to further the understanding of relevant changes in immune cell numbers and function during transfusion (and ICS). Information gathered from this study include alternative methods (never used before in TRIM research) and changes that occur peri-operatively. In future study, while working towards the clarification of TRIM, some of the specific changes differentially altered (when considering transfusion groups) in our study should be considered.”

3. Reviewer 1 comment: “As the authors claimed, the change of immune status during the peri-operative period could be affected by many factors. In this research, only elective orthopaedic cases were recruited. However, the detailed clinical information was missing. Were there differences, like bleeding volume, operation time and scope, etc.，between these groups? Did the patients who received ICS or ICS&RBC suffered more trauma or bleeding during the operation? In addition, would the patients show similar change of immune status when receiving other surgeries?”

We agree with reviewer 1 that a group of similar procedures were considered. This was intentional to be able to compare: not different immune responses between different surgeries, but indeed between various transfusion types/modalities. We agree with reviewer 1, assessments of these changes in other surgical procedures may be valuable in future but are beyond the scope of this study.

All the patients in our study were booked for ICS because they were scheduled to receive major elective surgery with the potential for major blood loss. These were the inclusion criteria for the study and usually for the ICS booking process at the RBWH. The group with ‘no transfusion’ relates to those where transfusion was not required (i.e., major orthopaedic surgery where transfusion was not required). So agreed, they were different in this aspect; they did not need transfusion. Some patients in our study received transfusion because they experienced major blood loss; while others did not require transfusion because they did not experience major blood loss. To reflect on the fact that major blood loss occurred peri-operatively we included the peri-operative changes in haematocrit in figure 2. 

We agree that all the factors mentioned by reviewer 1 (“like bleeding volume, operation time and scope”, and many other considerations) would be relevant in a clinical outcome study that aim to confirm ICS advantages over RBC. The aim of our study was not to evaluate if ICS is better than ABT, or to explain why patients receive ICS or RBC, but instead to observe what changes occur in biology when patients experience a large amount of blood loss during orthopaedic surgery and receive different transfusion modalities.

The group that had ‘no transfusion’ had similar surgery and anaesthesia but no transfusion and immune cell numbers did not change significantly over the surgical time. If certain immune cell populations did not change during peri-operative transfusion, they will likely not be of value to use in future TRIM research; or in studies aiming to assess associations with clinical outcomes. However, in the transfused groups (ICS and ICS&RBC) (those with major blood loss) immune cell numbers did change significantly. When using specific biomarkers (demonstrated to be differentially altered) in a larger study (when adequately powered for clinical outcomes) it will enable the consideration of all these other factors. But in this preliminary study we can only consider biological changes. 

Added to the manuscript: “Surgeons at the RBWH, expecting potential major blood loss during specific procedures, request the availability of ICS (booking). The RBWH cares only for patients older than 12 years of age. Within the observational nature of this study, ongoing recruitment of major elective orthopaedic surgical procedures to the study occurred. There were no exclusion criteria and no patients declined consent. At the RBWH ICS is now part of the standard practise for all elective orthopaedic procedures with major blood loss risk. Blood samples were collected from the arterial line (already in place for the surgery) before and after surgery and by using a standard phlebotomy technique at 48h. This study was observational, no changes to the clinical care of patients were made for the purposes of this study. None of the patients in this study had contraindications to use ICS. Reinfusion of ICS blood was only dictated (as clinically relevant) by the anaesthetist caring for the patient at the time. The sample sizes were based on previous experience of the authors in this manuscript, of similar preliminary studies.”

4. Reviewer 1 comment: “For the results of all patients and patients divided by groups, there was an obvious inconsistency. Patients without transfusion did not show significant change of immune cell populations and sub-populations, while those with transfusion did. Due to the limited number of patients without transfusion, the result of all patients showed difference. This conclusion was not rigorous. If recruiting more patients without transfusion, the result of all patients might be opposite.”

We agree, we had different observations depending on whether all the study patients were considered together or in divided groups. The significant differences in “all the patients” graphs were likely due to changes in the transfused groups. We did anticipate different results for the transfused groups. Therefore, while we started to assess overall changes (all patients included) to develop context (” bigger picture”), we then considered each study group individually (separate analysis). The biomarkers that stood out as being differentially modified in the study group analysis are likely the biomarkers of most value to use in future clinical outcome studies.

Agreed, the significant differences in “all the patients” graphs were likely due to changes in “transfused” study groups. For example, despite the small number in the ICS&RBC group (n=4), significant changes were seen.

Added to the manuscript: “The significant differences in “all the patients” graphs were likely due to changes in the transfused groups.”

5. Reviewer 1 comment: “The number of patients recruited was 19 showed in the Abstract. However, in the part of Patient recruitment, there was 5 patients of no transfusion, 8 patients of ICS, 4 patients of ICS&RBC and 1 patient excluded (5+8+4+1=18). Moreover, in the figures, there were 4 dots for the no transfusion group and 9 dots for the ICS group.”

Thank you to Reviewer 1 for identifying this error. We will correct this. While all figures and analysis included the correct number of analysed patients in the original manuscript, the section “Materials and Methods” reflected incorrect sample sizes.

We deleted from page 8: for the Materials and Methods section: “Responses were compared across the three study transfusion groups: those who had 1) no transfusion (n=5), 2) ICS only (ICS, n=8) and 3) both ICS and RBC (ICS&RBC, n=4)”. 

I copy here the correction (track changes include the relevant correction): “One patient was excluded for whom we did not have a 48h sample (the patient was discharged from hospital before the sample was due). Responses were compared across the three study groups: those who had 1) no transfusion (n=4), 2) ICS only (ICS, n=9) and 3) both ICS and RBC (ICS&RBC, n=4) (total number of patients analysed were n=17).”

Reviewer 1 comment: “Minor concerns:

1.No FCM gating figure or gating strategy of the result.”

Find added to manuscript: “Gating strategy was based on a previous study of cardiac surgery patients [6].”

2. ”In the figures, the arrow on the left indicated that the ordinate was measured in 10^9 cells/L. However, the results of cell populations measured in percentage were mixed. In addition, for cell sub-populations in figure 4 and 5, the units for the cell account should be mistake.”

Thank you. Please see now corrected in figures: “Cells 109/L” (Fig 2 and Fig 3) and “Cells/µL” (Fig 4 and Fig 5).

3. “Regulatory T cells were defined as CD4+CD25+CD127- cells in the part of Materials and methods. However, in the part of Results, CD4+CD25+ T cells were regarded as Tregs (Line 260, Page 13).”

Please find the following changes (lines 260, 264, and 298): we deleted “regulatory” from “In our study regulatory CD4+ CD25+ T cells”, changed to “In our study CD4+ CD25+ T cells”.

4.”The FCM channel for detection of Co-stimulatory/Adhesion Molecules and pDC was unknown.”

The fluorochrome used was PE, which was assessed using a blue laser and a 585 nm band pass filter. The table of antibodies has been updated to reflect PE was used.

Reviewer 2 comments:

Reviewer #2: "I am a clinician with experience of ICS and welcome that research is being performed in this area. Concerns relating to TRIM are hugely important though knowledge of mechanism and risk are not well known. I see this very detailed work has been done to shed light on the topic and raise further questions and identify topics for investigation. I found it hugely complicated and confusing to understand what the actual effects were of the intervention and the potential effect upon immune function as a result. I note the n of each group is small and then there is the problem that any issue found from ABT (which is the "alternative" to ICS) is confused by the fact that the ABT group contains ICS. The TRIM issue is suspected to be dose dependent (Horvath et al - donor transfusion and peri-op infection ). I could not identify whether the "dose or volume" of transfusion product was considered."

We agree with reviewer 2 that “Concerns relating to TRIM are hugely important though knowledge of mechanism and risk are not well known”. 

We consider each important aspect from this paragraph separately here below:

Reviewer 2 comment: “I found it hugely complicated and confusing to understand what the actual effects were of the intervention and the potential effect upon immune function as a result.”

Thank you to reviewer 2 for the opportunity to clarify this very important aspect of our study. This manuscript aims to identify changes in immune responses during transfusion. We do not aim to give a final answer of what benefit ICS has upon immune function. ICS was not considered as an “intervention” in this study. ICS was previously associated with improved outcomes in clinical studies. However, we know very little about immune responses during ICS. In this study we found changes in immune cell populations and sub-populations (the panel was specifically chosen to use when considering specific clinical outcomes in future study) specifically altered following ICS and ICS&RBC. These changes did not occur in the “no transfusion” group. Whether the changes we identified would be advantageous or not (during clinical outcome studies) remains to be seen. This can only be confirmed in a large clinical outcome trial. But now we know which biomarkers (in terms of changes in immune cell numbers) would be of value to include in such a trial.

Reviewer 2 comment: “I note the n of each group is small …”

We agree the sample sizes were small. These small sample sizes made the study feasible. Because of the small sample sizes, we could assess a large panel of biomarkers, providing a better position to understand simultaneous changes for many biomarkers. 

Our study succeeded in the assessment of a large panel of immune cells and demonstrated significant changes in immune cell sub-populations. We found many significant but complex changes during surgery and transfusion (despite the small sample sizes). 

A refined panel of “biomarkers” (i.e., those identified as differentially altered) would be most valuable to use in future study. These biomarkers can now be applied to a larger clinical study to evaluate clinical outcomes, specifically related to each relevant result in our study.

I copy from manuscript (discussion): “While the small sample size is a limitation of the conclusions that can be drawn, we have established in principle that numbers of immune cells changed peri-operatively, and these changes were specifically associated with different transfusion modalities.”

Reviewer 2 comment: "…and then there is the problem that any issue found from ABT ( which is the "alternative" to ICS ) is confused by the fact that the ABT group contains ICS.”

and

“…I would have considered that there needed to be three distinct groups ( control, ABT, ICS) and hence the effects are more clear as to whether it is transfusion alone or donor blood or ICS that affects the white cell populations.”

And

“…I also think that an ICS only and Transfusion Only group would be required, but understand the ethical issues behind this ( if there is a C/I for using ICS, this may already echo a difference in the immune status of the patient).” 

While this study was an observational study (with no changes to clinical care), the study groups were not dictated by the study, but depended on the clinical opinion of the attending anaesthetist. We included relevant elective surgical procedures at an ongoing basis within the study period (with no advice to the treating teams to use ICS, or any other changes to clinical care). There were no exclusions due to contraindications (C/I). These groups reflect the most common clinical requirements now, and likely for the future (i.e., ICS or ICS&RBC). 

We agree, if this study did aim to assess the value of ICS over ABT, it would require an ICS and RBC group (i.e., ICS, RBC, control). This was not the aim. This may also not be possible in future; seeing ICS is available and widely considered beneficial for this cohort at our institution (and internationally). Recruitment to a “no ICS” group may not be considered ethical during these major orthopaedic procedures anymore.

Instead, this study aims to clarify changes that occur during transfusion (in groups that are currently clinically relevant). We know through extensive literature review that ICS has certain benefits, for example when it comes to infection risk. We do not know where to start if we want to consider biomarkers to assess in a trial where ICS and allogeneic/donated blood transfusion (RBC or ICS&RBC) are compared. Therefore, we took this large panel of biomarkers and evaluated which changes occurred. In this way we can now say that changes identified within this manuscript may be important to consider in a large clinical outcome trial.

Reviewer 2 comment: “The TRIM issue is suspected to be dose dependent ( Horvath et al - donor transfusion and peri-op infection ). I could not identify whether the "dose or volume" of transfusion product was considered.”

Please also see our earlier reply considering confounding factors (i.e., “volume blood loss”, “dose of blood products transfused” etc.)

We considered a large panel of “biomarkers”. This was possible because of the smaller sample size in the study. We found many significant but complex changes, and some biomarkers that did not change. Now that we know which biomarkers are of most value in these circumstances, a subsequent clinical outcome study with adequate power could consider confounding factors such as "dose or volume" of transfusion.

Reviewer 2 comment: “I tried hard with multiple re-reads to really understand what was going on and what the inference or points were that the authors wish to portray to the reader…. For the clinician, does the work give a clear indication that ICS has an immunological effect, should patients be informed that TRIM occurs with ICS , is this different to the TRIM of donor blood?”

As a clinician myself I agree with reviewer 2, the biomarkers in this study represent complex immunological changes. It is not possible to relate these changes to direct patient outcomes (or to use results in this manuscript when providing information to patients yet). Changes in immune cell sub-population numbers have never been studied in this way when considering TRIM (or ICS). This is novel preliminary work.

Each immune cell sub-population (each represented in a graph in figure 5) was chosen, 1) due to its relevant “function” (i.e., role within the immune response) and 2) its potential association with clinical outcomes within allogeneic transfusion literature. Each significant difference identified in our study is a “message” (key finding), and each difference (considering transfusion groups) is a potential avenue to consider during future TRIM research. 

Standard methods, previously used to find the cause/ describe the mechanism of TRIM, has so far not provided all the answers. This is the first study of its kind. We identify important peri-operative changes in immune cell responses, never identified before. Furthermore, we identify a new approach, a new method to potentially use in future when studying the potential causes of TRIM; when (in future) studying peri-operative adverse outcomes following transfusion (specifically after ‘orthopaedic surgery that require transfusion’), and when comparing ICS and RBC (or ICS&RBC) transfusion in the future.

We do not have all the answers yet, this work describes the progress towards finding answers. We are one step closer to addressing this point (i.e., “give a clear indication that ICS has an immunological effect “). As a clinician myself, I did a thorough investigation (the reason for the extensive literature review), to consider the clinical benefits that each change we identified, may present to our patients. I hope that we can apply information gained from this elegant cutting edge (but time consuming) immunological study to a larger clinical trial in future. Therefore, by using a smaller panel of biomarkers (those differentially modified following transfusion) it may become feasible to consider clinical outcomes in patients following ICS.

We cannot draw clinical conclusions to provide to patients when talking about the benefits of ICS over ABT from this study, yet. We can only conclude that changes occur during transfusion, these changes are different considering ICS and ICS&RBC and may potentially be valuable when studying the mechanism of TRIM in future.

Our message is (I copy from the conclusion): “When considering all the study patients, peri-operative changes in immune cell populations and sub-populations were significant. There were no significant peri-operative changes, for all immune cell populations studied (cell numbers and proportions (%)), in surgical patients who did not receive transfusion. However, when considering transfused groups (ICS&RBC and ICS), significant changes were evident in population numbers and functional capacity (e.g., co-stimulation markers, adhesion, activation, and regulation). Areas of interest that would be valuable to study when comparing ICS vs RBC in future include: CD4+ CD25+, CD4+ CD25-, CD4+ CD127+ and CD25+ CD127- T cells, DC cell numbers (CD11+ cDC), capacity to respond (CD9+ cDC) and pDCs. Further study of changes in immune cell sub-populations, within the peri-operative journey, and in association with different transfusion modalities, may provide valuable insights in the evaluation and prediction of infection related adverse outcomes, peri-operatively in the future [7].”

Reviewer 2 comment: “Would a benefit in message and ease of interpretation be achieved if the number of cell types were reduced in this paper, even considering a second paper to describe additional cell types.”

ICS has previously been demonstrated as beneficial when considering clinical outcome studies. Yet there is a gap in the literature to inform us about which biomarkers would be of most value during a clinical outcome study. We specifically chose this panel of immune cells to direct future study towards potentially associated clinical outcomes (identified through extensive literature review). Each individual sub-population studied considered a specific aspect of the immune response (“specialised immune sub-populations”), and a specific association with clinical outcomes (“could be used as cell targets in larger clinical outcome studies”).

I quote from the original manuscript: “We first considered the routinely assessed populations identified in a basic differential full blood count and further studied an extended panel of specialised immune sub-populations, using quantitative flow cytometry. We used this approach to identify immune sub-populations modified by transfusion that could be used as cell targets in larger clinical outcome studies.”

Specific biomarkers were demonstrated to be differentially altered following different transfusion modalities (i.e., ICS, ICS&RBC, not transfusion). Using the large panel of biomarkers in our study is not realistic when studying a large enough population to consider outcomes for example infection. So, this preliminarily work was essential to “narrow this down” (to a refined panel). Previously used “biomarkers” have not provided clarification when studying the mechanism of TRIM and the pathophysiology behind the improved outcomes following ICS. We now present a new option. We can now use a much smaller biomarker panel (not previously identified as differentially modulated) when we compare these “types” of transfusion (transfusion modalities) in future.

Reviewer 2 comment:

Specific remarks

42- received- spelt incorrectly

Our apologies, we did update the manuscript to include this correction.

288 - unit 109/L - unsure what this refers to

Our apologies, we did update the manuscript to include correction: 

Corrected to 109/L, thank you.

287-297 I could not understand this section.

This section, lines 287 to 297, contains the legend for figure 5. We will present the figure heading in bold and include a space to make this clearer.

307- in the results section, this sentence appeared to be out of context and such a phrase should have been either in the introduction of discussion section.

We acknowledge that the information for the DC cell is included for context. We acknowledge we have studied many immune cells and were just reminding the readers of their role. We added: “DCs were assessed within our panel, because they…”

Reviewer 2 comment: “Clearly a huge amount of very focused and specialised study and extensively researched discussion. But, due to the complexity, I fear the message and inference/influence may be obscured. I would find it difficult to include a "message " from this paper in any presentations or papers that I would write on this subject, and this is probably due to the complexity and comprehensive nature of the text.”

We acknowledge it is a complex and comprehensive study. We have taken steps to refine the core information, to improve readability and ensure the key outcomes are clearer. Please find therefore relevant changes within the revised manuscript and a major rewrite of the discussion section, with the aim to refine the purpose of the study and key messages. Additional literature references were removed.

---

## [Editor Report · Decision Letter 1]

13 Jul 2023

Intraoperative cell salvage: The impact on immune cell numbers

PONE-D-23-10523R1

Dear Dr. Roets,

We’re pleased to inform you that your manuscript has been judged scientifically suitable for publication and will be formally accepted for publication once it meets all outstanding technical requirements.

Kind regards,

Afsheen Raza, PhD

Academic Editor

PLOS ONE
---

## [Editor Report · Acceptance letter]

19 Jul 2023

PONE-D-23-10523R1 

Intraoperative cell salvage: The impact on immune cell numbers 

Dear Dr. Roets:

I'm pleased to inform you that your manuscript has been deemed suitable for publication in PLOS ONE. Congratulations! Your manuscript is now with our production department. 

Kind regards, 

on behalf of

Dr. Afsheen Raza 

Academic Editor

PLOS ONE